# Achieving sub-pm wavelength regression via minimum-phase in a single-stream photonic IC

Hector A. Rubio Rivera ⬤ , Lilian Neim, Venkatesh Deenadayalan & Stefan Preble ✉

Photonic chips are powerful tools for measuring and analyzing light, but most compact spectrometers face a fundamental trade-off: improving resolution usually requires larger devices or sacrifices in signal quality. Here, we introduce a chip-scale architecture that overcomes this limitation by extracting phase information corresponding to the hidden timing of light waves using only simple intensity measurements. Our method generalizes earlier minimum phase designs to allow sparse and non-sequential optical delays, enabling accurate phase reconstruction on a single circuit. By engineering these delays, the device can determine the wavelength of an unknown laser with sub-picometer precision, all while using just one input and one output. This single-stream design reduces loss, improves robustness, and avoids the complexity of traditional spectrometers. The result is a compact, scalable platform that enables high-accuracy wavelength metrology and opens possibilities for on-chip sensing and computational spectroscopy.

The field of computational spectroscopy has recently received a great deal of attention in the photonic integrated circuit research community[1-3]. The main reason behind this is the ability to incorporate functionalities that, in the past, required large footprints[4-10]. Photonic integrated circuits (PICs) allow the miniaturization of all these bulky systems while maintaining their functionality and desired performance[11]. In addition to the size advantage, the ability to mass-produce these miniaturized spectrometers enables a more cost-effective solution for the consumer market.

PIC spectrometers have been implemented using three primary architectures: Mach-Zehnder interferometers (MZIs) with multiple optical path lengths[12-21], optical cavities that partition the spectrum into discrete channels[22-27], and dispersive elements that spatially spread the spectrum for reconstruction[28-33]. Each offers distinct strengths but faces fundamental trade-offs between resolution, bandwidth, footprint, and signal fidelity. Dispersive designs can access a broad bandwidth, but they require many spatial channels, which increases size and loss. Optical cavity spectrometers achieve high resolution with large quality factors, yet wide-band operation demands multiple cavities. MZI-based approaches are broadband, but

resolution scales directly with path length, and intensity-only regression near interferometric nulls produces error bias. In all three cases, the multi-path nature of the architecture inherently reduces SNR, as optical power is divided among multiple channels or paths.

Using the light's phase overcomes the drawbacks of interferometric fringes for spectroscopy. However, accessing the phase information in a photonic circuit is nontrivial. For example, it is common to utilize integrated 90° hybrids[13], but they require a large footprint and are highly susceptible to manufacturing variations. In contrast, Xu et al. showed that the phase in a photonic circuit can be extracted solely from intensity measurements using minimum-phase systems[34,35]. Specifically, the phase is determined directly from the magnitude via their fundamental relationship in a minimum phase system, enforced by the Hilbert transform[36-38].

Here we generalize the work of Xu et al., who applied the minimum phase concept specifically to optical FIR (Finite-Impulse Response) filter implementations[34]. Our generalization extends the applicability of the minimum-phase approach to high-performance wavelength metrology and spectroscopy applications, as well as to general circuit parametric extraction (key for the development of PIC

Electrical and Microelectronic Engineering, Rochester Institute of Technology, Rochester, NY, USA. ✉e-mail: sfpeen@rit.edu

process design kits, an area of growing commercial importance). Specifically, Xu et al. demonstrated that the full complex response of a circuit can be reconstructed from intensity-only measurements by leveraging the minimum-phase Kramers-Kronig relationship[36]. However, their implementation was limited to sequential tap filter configurations ($n = 1,2,3,...$). This conceptual limitation would prohibitively increase a spectrometer circuit's footprint by n-fold (n delays paths needed to go from bandwidth-defining FSR to higher-resolution FSR). Here, we overcome this through the use of non-sequential tap increases, uncovering the necessary optical signal processing needed to achieve high-performance wavelength metrology. This work enables arbitrary and sparse tap filter configurations, where n can be any integer, removing the footprint requirements from n delay paths to any $m < n$. Enabling this arbitrary increase in delay length significantly enhances spectral resolution, proving particularly valuable for wavelength metrology applications. Specifically, in this work, we utilize large tap values to achieve sub-picometer (sub-pm) wavelength resolution in a single-stream circuit. As a result, our approach overcomes the fundamental trade-offs faced by traditional multi-path architectures, which must balance resolution, footprint, and signal/noise ratio.

In this work, we present a single-stream, minimum-phase photonic wavemeter that reconstructs optical phase from intensity measurements using a sparse multi-$\Delta L$ architecture, enabling absolute wavelength inference with sub-picometer precision over user-defined windows. We derive the design rules that ensure minimum-phase operation in asymmetric Mach-Zehnder interferometers, and we show how a zero-phase FIR approach isolates the relevant delays while preserving phase fidelity. We develop wavelength-regression models that use the uniquely determined short-delay phase to order longer

delays, and we incorporate a second-order dispersion term for high-accuracy inference at the largest $\Delta L$. We validate the approach on fabricated PICs, and we quantify error and robustness within practical bandwidth limits. Section "Minimum-phase Mach-Zehnder interferometer" details the theory and design; Section "Single $\Delta L$ Design" presents the wavelength-regression methods for single and multi-$\Delta L$ regression methods; Section "Sparse filter structure for custom wavelength windows" demonstrate absolute wavelength retrieval across user-defined windows with sub-pm error; Section "Discussion" discusses limitations and future work; and Section "Methods" describes fabrication and measurement.

## Results

### Minimum-phase Mach-Zehnder interferometer

We analyze the minimum phase condition for an integrated MZI by examining its transfer function from the schematic presented in Fig. 1(a). The transfer matrix method is used to derive the MZI transfer function given in Equation (1).

$$H(\omega) = \frac{Y_1}{X_1} = \frac{\tau}{\sqrt{2}} e^{-\alpha L_1} e^{-j\beta L_1} - \frac{\kappa}{\sqrt{2}} e^{-\alpha L_2} e^{-j\beta L_2} \qquad (1)$$

where $\kappa$ and $\tau$ are the input coupler coefficients and are related by $|\tau|^2 + |\kappa|^2 = 1$, $L_1$ and $L_2$ are the MZI arm lengths, $\beta$ is the wavevector and is given by $\beta = \frac{2\pi n_{eff}}{\lambda}$ with $\lambda$ and $n_{eff}$ being wavelength and the waveguide effective index, and $\alpha$ being the waveguide loss per unit length.

Following the convention defined in Supplement, Section 1, we let $H_{ref}(\omega) = \frac{\tau}{\sqrt{2}} e^{-\alpha L_1} e^{-j\beta L_1}$ and $H_{cpu}(\omega) = \frac{\kappa}{\sqrt{2}} e^{-\alpha L_2} e^{-j\beta L_2}$, where $L_2 > L_1$, and we have assumed that the directional couplers are broadband in the wavelength window of interest. Using these substitutions, the MZI

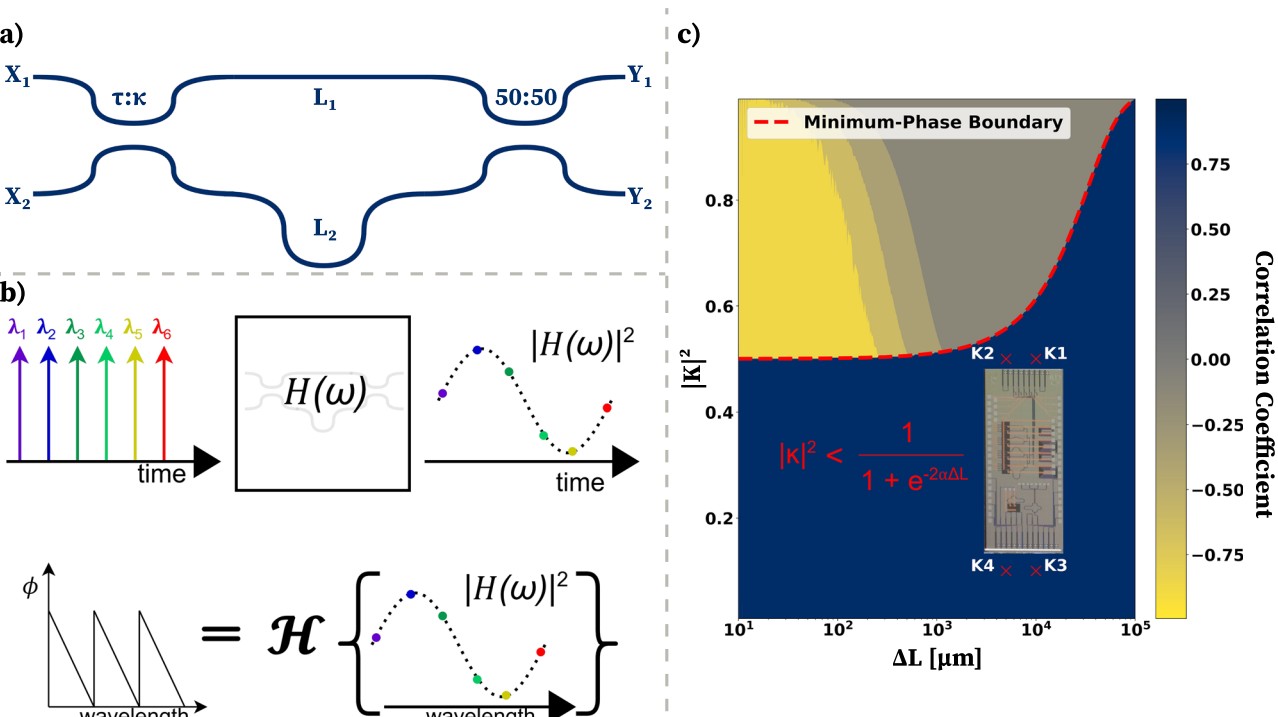

**Fig. 1 | Minimum-phase Mach-Zehnder inteferometer for phase retrieval from intensity measurements. a** Asymmetric Mach-Zehnder Interferometer(MZI) with a variable input coupling coefficient $\kappa$ and a 50: 50 output coupler used to analyze the minimum-phase boundary condition. **b** Phase retrieval process using a wavelength-swept laser, where the $H(\omega)$ intensity response is related to the phase response through a Hilbert transformation owing to the minimum-phase nature of the system. **c** The simulated correlation coefficient between the

retrieved phase (by Hilbert transform) and the simulated phase shows excellent agreement with the derived minimum-phase boundary condition (red-dashed line). All points in the blue region meet the minimum-phase condition. The red X marks inside the contour plot show the chosen four design points used to test this condition experimentally using a PIC (microscope image). K1 and K2 are near the minimum-phase boundary and K3 and K4 strongly meet the minimum-phase condition.

becomes a minimum phase system when $|\frac{-H_{cpu}(\omega)}{H_{ref}(\omega)}| < 1$, which provides us with the minimum phase boundary for an MZI, as given in Eq. (2)

$$\left| \frac{-H_{cpu}(\omega)}{H_{ref}(\omega)} \right| < 1 \Rightarrow |\kappa|^2 < \frac{1}{1 + e^{-2\alpha\Delta L}} \tag{2}$$

where $\Delta L = L_2 - L_1$. This means that, to retrieve the phase of the MZI transfer function, the input coupling coefficient ought to be smaller than the sigmoid dependence on the optical path length difference of the two arms. As a result, if the input coupling coefficient meets this condition, we can retrieve the phase of the photo-detected intensity relationship ($G(\omega)$) of the MZI shown by Eq. (3).

$$H(\omega) = e^{-j\beta L_1} e^{-\alpha L_1} \left( \frac{\tau}{\sqrt{2}} - \frac{\kappa}{\sqrt{2}} e^{-\alpha\Delta L} e^{-j\beta\Delta L} \right)$$
$$= e^{-j\beta L_1} G(\omega) \tag{3}$$

where $G(\omega)$ has a constant DC factor ($\frac{\tau}{\sqrt{2}}$) and a linear phase ($e^{-j\beta\Delta L}$) sans dispersion. We note that $G(\omega)$ is the photo-detected signal that is measured at the MZI output. Inspecting Equation (3) shows that the relationship between $G(\omega)$ and $H(\omega)$ is the global phase offset $e^{-j\beta L_1}$ that is lost due to photodetection being a complex magnitude operation.

We perform a simulation to validate the minimum-phase boundary condition (Eq. (2)) by varying the values of $\Delta L$ and $\kappa$ of the MZI. As described in Fig. 1b, the phase of $H(\omega)$ is recoverable from its intensity relationship in a given bandwidth when $H(\omega)$ is a minimum-phase system. As a result, we compare the retrieved phase given by

$$\theta(\omega) = \mathcal{H}\{\log(|G(\omega)|)\} \tag{4}$$

where $\theta(\omega)$ is the $G(\omega)$ phase, and $|G(\omega)|$ is the intensity of $G(\omega)$, with the simulated phase $arg(G(\omega))$ as a function of both $\Delta L$ and $|\kappa|^2$. In order to validate the conditions under which minimum-phase occurs in Fig. 1c, we use the Pearson correlation coefficient to compare the extracted phase with the simulated phase. The correlation coefficient is maximized when the minimum phase boundary condition is met, as shown by Fig. 1c (blue shaded region below the minimum phase boundary). Furthermore, we note that the correlation coefficient approaches $-1$ when $\Delta L$ is small and $|\kappa|^2 > 0.5$. This happens because the partitioned paths are similar in length, causing $\kappa$ to present a stronger minimum-phase condition than $\Delta L$. As a result, the roles of reference and delay paths are reversed in this region, causing a phase reversal, i.e., a multiplication by $-1$, shown by the negative correlation coefficient. Consequently, we conclude that with MZI's meeting the minimum-phase condition (Eq. 2), we can successfully recover the phase of $G(\omega)$ by using Eq. (4).

## Single $\Delta L$ design

**Algorithm 1**. Single $\Delta L$ Wavelength regression algorithm.
**Require**: wavelength ($\lambda$), and photo-detected power ($P$) from calibration laser S1. Photodetected signal $\hat{P}$ from test laser S2.

```
1:  function EXTRACTG (P)                    ▷ Extract G(ω) from P
2:      φ ← H{lnP}
3:      h(n) ← F⁻¹{√P e^{jφ}}
4:      G ← F{h(n) × w_min(n)}
5:      return G
6:  function CALIBRATION (λ, P)              ▷ Model for λ regression
7:      G ← ExtractG(P)                      ▷ obtain e^{jβΔL} response
8:      n_peak ← find_peaks (|F⁻¹{G}|)
9:      h_filter ← F{w_bpf(n)}               ▷ Design band-pass filter
10:     H_filtered ← h_filter*G_copies       ▷ apply filter
11:     θ ← arg(H_filtered)                  ▷ Linear phase
12:     C₁, C₀ ← polyfit(Δf, unwrap(θ), 1)
13:     return C₀, C₁
14: function TESTING (P̂)                     ▷ wavelength inference
15:     G ← ExtractG(P̂)                      ▷ obtain e^{jβΔL} response
16:     Ĥ_filtered ← h_filter*Ĝ_copies       ▷ apply designed filter
17:     θ̂ ← arg(Ĥ_filtered)                  ▷ Linear phase
18:     Δf ← C₁(θ̂ − C₀)                      ▷ Apply model
19:     λ̂ ← c/(Δf+f₀)                        ▷ c:= speed of light
```

Here we demonstrate wavelength regression using the experimental process and the wavelength regression algorithm shown in Fig. 2 and Algorithm 1, respectively. A PIC with four different MZI circuits with varying input coupling coefficients (90:10 and 50:50), and $\Delta L$ (0.5 and 1 cm) was designed to experimentally test the boundary condition of the minimum phase as shown in Fig. 2a and indicated in Fig. 1c. The wavelength regression algorithm is based on calibrating the wavemeter and then testing it using an unknown laser to validate the prior calibration. The output of the MZI is directly connected to an on-chip photodetector, and we read the photo-generated current as a function of the MZI's input wavelength, given by either a known value from the calibration laser (S1) or an unknown value given by the uncalibrated semiconductor tunable laser diode (S2). To assess the performance variance from chip-to-chip, two packaged PICs (P1 and P2− one of the packages is shown in Fig. 2a) are utilized to cross-correlate training and testing data.

The intermediate steps of the wavelength regression algorithm are shown in Fig. 2. The algorithm starts by extracting the phase ($\phi$) from the MZI's intensity response by applying the *Hilbert transform* to the intensity ($P$). Subsequently, the discrete-time complex-valued response ($h(n)$) is constructed by taking the inverse *Fourier transform* of the *constructed* complex-valued frequency response. Finally, $G(\omega)$ is obtained by eliminating the $n = 0$ tap by using a window function ($w_{min}(n)$) that is zero at the $n = 0$ point and one everywhere else. After obtaining $G(\omega)$, which corresponds to the $e^{j\beta\Delta L}$ information of the circuit, we use SciPy find_peaks algorithm to find the peak present in the Fourier transform of $e^{j\beta\Delta L}$[39]. Using the peak information, we define a *band-pass filter* ($w_{bpf}(n)$) centered around this peak. Subsequently, the filter is used to eliminate any spectral components that are not the linear phase found in $G(\omega)$, as shown by *the apply filter stage*. The phase of the filtered response is then *unwrapped and* used as the basis for linear regression using Numpy's polyfit function[40] that produces the linear model coefficients $C_0$ and $C_1$ used for wavelength inference. To perform wavelength inference, we need to find the roots of our chosen polynomial function (linear in this case) and then convert the result into wavelength. We use frequency as our regression axis because the wavevector $\beta$ expansion as a function of frequency is a more tractable function, as shown in Eq. (5). While we expand $\beta$ to the second term in the Taylor series ($\propto \Delta\omega^2$), we only use the first term for the results presented in this section (the linear model). We will use the second term in the next section due to a need for modeling dispersion at large $\Delta L$. Note that our results are all shown in wavelength by using the relation $\lambda = c/f$ where $c$ is the speed of light.

$$\beta = \frac{\omega n}{c} \approx \frac{\omega_0 n_0}{c} + \frac{n_g}{c}\Delta\omega + \frac{1}{c}\frac{dn_g}{d\omega}\Delta\omega^2 \tag{5}$$

where $n_g$ is the material's group index and $\beta$ was Taylor expanded around $\omega_0$.

The discrete-time filter response is designed to be real, implemented as a window of length $2+1$ centered on the peak of $H(n)$, where $m$ is an integer. The filter tap coefficients are then obtained via a Fourier transform, yielding a zero-phase filter[41]. This approach prevents phase distortion of the experimental data during the convolution operation $h_{filter}*G_{copies}$. Importantly, the convolution is performed with $G_{copies}$

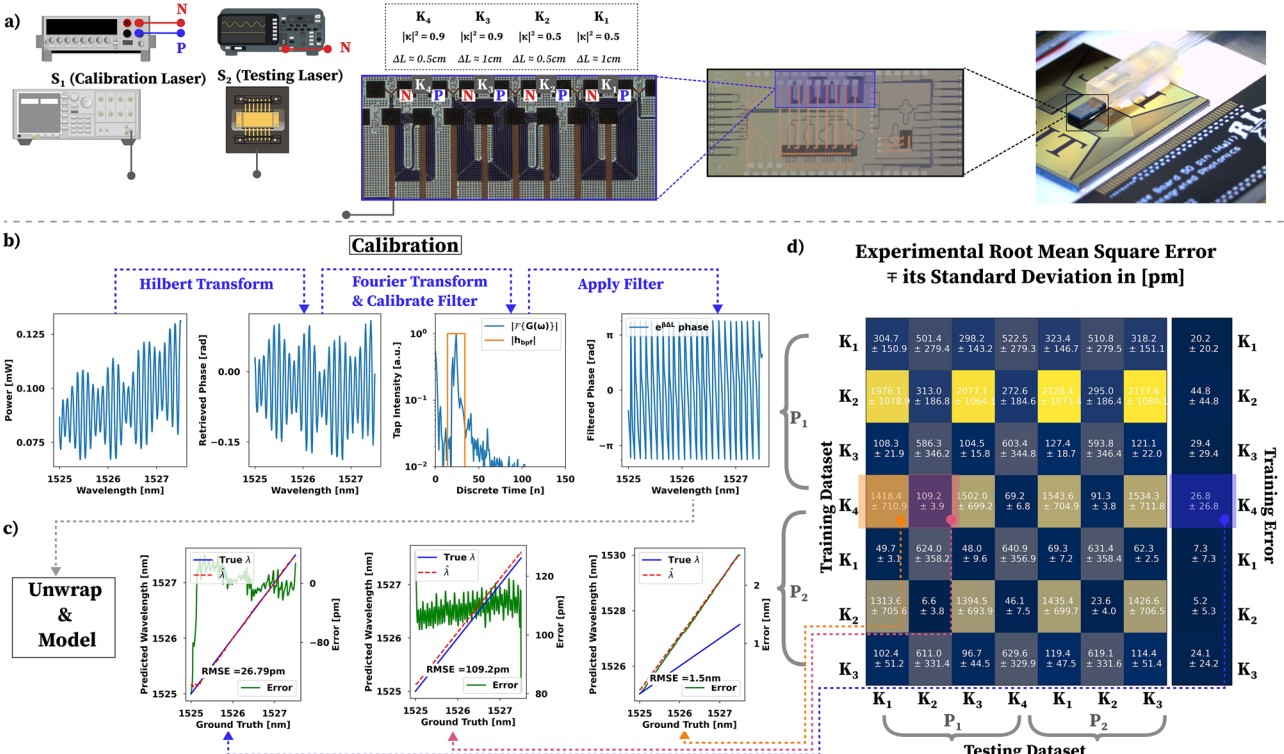

**Fig. 2 | Wavelength regression with minimum-phase MZIs. a** Experimental setup used to characterize the four minimum-phase MZI PIC designs (K1, K2, K3, and K4). The PIC was packaged to allow for robust and straightforward wavelength regression tests. The experimental setup uses a calibrated Keysight 81606A tunable laser source (S1) to calibrate the PIC circuits. The testing setup uses an unknown continuously swept tunable semiconductor laser (S2). The 4 MZIs are designed using a directional coupler, with two different coupling coefficients (50:50 for K1 and K2 and 90:10 for K3 and K4), as the input coupler and a Y-splitter as the output coupler. The outputs are read out through wirebonded photodetectors with a Keithley 2400 source-measure unit (SMU) and an oscilloscope. **b** Depiction of the algorithm used for wavelength regression showing the filter design process and the resulting filtered phase. **c** Results for the wavelength regression with the filtered phase used as the regression basis. **d** Furthermore, the matrix shows the Root Mean Squared Error (RMSE) for both calibration and testing procedures for two packaged chips ($P_1$ and $P_2$) calibrated by the row circuit and tested by the column circuit.

rather than the original $G(\omega)$ because filtering is a linear convolution process sensitive to filter length. When the filter exceeds the data length, its transient response introduces phase artifacts. To mitigate this, multiple copies of $G(\omega)$ are concatenated to buffer the filter application, and the transient regions are subsequently discarded.

To assess the algorithm's validity, we compare different training and testing datasets. As an example, the plots in Fig. 2b show the algorithm applied to the experimental data for package $P1$ and circuit $K4$ for the calibration step and data from $P1/K1$ and $P1/K2$ for the testing of the regression results in Fig. 2c. More importantly, we show what happens in the algorithm when we test matched $\Delta L$ (test data from $K_1$ with calibration data from $K_4$—red arrow) and mismatched $\Delta L$ (test data from $K_3$ with calibration data from $K_4$—orange arrow). This data shows that when the $\Delta L$ values of the circuits are matched (red arrow), the error is minimized as the retrieved phase slopes are the same. In contrast, when the $\Delta L$s are mismatched, the error increases due to phase slope mismatch (orange arrow). The full characterization of the two packaged chips was carried out, and the resulting RMSE for each calibrating/testing step is shown in the matrix found in Fig. 2d. We see that the error is minimized when the $\Delta L$s are matched (blue squares), whereas the error increases when the phase slope is mismatched (yellow/gray squares). $P2−K2$ gives the minimum calibration RMSE of 5.2[$pm$], and the minimum testing RMSE of 6.6[$pm$] is given by the combination of training with $P2−K2$ and testing with $P1−K2$. This dataset proves that the experimental data drives the results, and the filtering process is not biasing our regression algorithm. More details regarding the application of the algorithm to matched and mismatched $\Delta L$ are presented in the Supplementary Section 2.

The regression presented here is sensitive to global phase shifts because the retrieved phase is not uniquely determined within the wavelength window of interest. Temperature variations will cause changes in the $y$-intercept in the filtered phase, leading to an RMSE bias. Further evidence of this effect is shown in Fig. 2c, where the matched $\Delta L$ testing data (red arrow) presents a small global phase offset when compared to the calibrated data (blue arrow), producing an RMSE increase. While we can account for these changes by minimizing the phase shift between the calibration and test data after filtering, this also introduces ambiguity in the retrieved wavelength, as the shift can also imply the wavelength window has changed. Therefore, in the next section, we present a circuit architecture that can handle these global phase shifts by using a delay path that produces uniquely determined phase data within the bandwidth of interest.

## Sparse filter structure for custom wavelength windows

The phase measurement of a single $\Delta L$ MZI design is not uniquely determined in the bandwidth of interest as $\beta\Delta L = \beta\Delta L + n2\pi$, where $n$ is any integer. Therefore, this approach is constrained to cases where the testing laser wavelength window is known a priori (such as laser stabilization). To circumvent this limitation, we designed a circuit that enables the recovery of a uniquely determined phase in the wavelength window of interest. To do this, we take advantage of the fact that the phase is directly proportional to $\Delta L$. As a result, we can define a value $\Delta L$ that will be uniquely determined in the operating bandwidth of the wavemeter. Furthermore, the addition of larger optical delay lines on top of the bandwidth-defining $\Delta L$ enables higher resolution measurements, achieving sub-pm resolution. The

resulting optical structure is that of a sparse optical filter where the optical delays are non-subsequent integer increases of the bandwidth defining $\Delta L$.

**Algorithm 2**. Multi-$\Delta L$ wavelength regression algorithm for absolute wavelength measurement.

**Require:** wavelength from known source ($\lambda$), photo-detected power from minimum phase MZI ($P$) in linear scale using known laser source.

1: **function** EXTRACTG ($P$)  ▷ Extract $G(\omega)$ from $P$
2:   $\phi \leftarrow \mathcal{H}\{\ln P\}$
3:   $h(n) \leftarrow \mathcal{F}^{-1}\{\sqrt{P}e^{j\phi}\}$
4:   $G \leftarrow \mathcal{F}\{h(n) \times w_{min}(n)\}$
5:   **return** $G$
6: **function** CALIBRATION ($\lambda, P$) ▷ Model for $\lambda$ regression
7:   $G \leftarrow ExtractG(P)$  ▷ obtain $\sum e^{j\beta \Delta L_n}$ response
8:   $n_{peaks} \leftarrow$ find_peaks ($|\mathcal{F}^{-1}\{G\}|$)
9:   $\Theta \leftarrow \{\}$
10:   $C \leftarrow \{\}$
11:   **for** $i \leftarrow 0$ **to** $len(N_{peaks})$ **do**
12:     $h_{filter}^i \leftarrow \mathcal{F}\{w_{bpf}^i(n)\}$  ▷ Design band-pass filter
13:     $H_{filtered}^i \leftarrow h_{filter}^i * G_{copies}$  ▷ apply filter
14:     $\theta^i \leftarrow \arg(H_{filtered})$  ▷ Linear phase
15:     $C_2^i, C_1^i, C_0^i \leftarrow$ polyfit($\Delta f$, unwrap($\theta^i$), 2)
16:     $\Theta \leftarrow + \{\theta^i\}$
17:     $C \leftarrow + \{C_2^i, C_1^i, C_0^i\}$
18:   **return** $C, \Theta$
19: **function** WAVELENGTHMODEL ($\lambda, P$)  ▷ $\lambda$ model
20:   **for** $i \leftarrow 0$ **to** $len(N_{peaks})$ **do**
21:     **if** $i == 0$ **then**
22:       $\Delta f \leftarrow \frac{-C_1^i + \sqrt{(C_1^i)^2 - 4C_2^i(C_0^i - \Theta[i])}}{2C_2^i}$
23:     **else** $FSR \leftarrow \frac{2\pi}{C_2^i \Delta f + C_1^i}$
24:
25:       $\delta f \leftarrow \frac{-C_1^i + \sqrt{(C_1^i)^2 - 4C_2^i(C_0^i - \Theta[i])}}{2C_2^i}$
26:       $\Delta m \leftarrow round(\frac{\Delta f - \delta f}{FSR})$
27:       $\Delta f \leftarrow \delta f + \Delta m FSR$
28:     **return** $\frac{c}{\Delta f + f_0}$  ▷ c: = speed of light

The reason why we need non-subsequent integer increases of the bandwidth defining $\Delta L$ is to increase the resolution of the final wavelength regression at the largest $\Delta L$ stage. It has been consistently shown in the literature that larger $\Delta L$ values allow for finer resolution[13,17,42]. Thus, we wish to incorporate this functionality into our design while maintaining a single-input-single-output architecture. As a result, all optical paths that form the distinct $\Delta L_i \forall i \in [1, 2, \ldots, N]$ paths need to be interferometrically coupled with each other at the output. This architecture resembles an FIR filter structure analyzed for filtering applications by Xu et al.[34]. However, in their analysis, the optical delay lines are assumed to be subsequent integer increases of the bandwidth-defining (smallest) $\Delta L$. Therefore, in the work presented here, this assumption has been eliminated in the analysis of this FIR filter structure, thereby generalizing that work. In this context, the resulting architecture is that of a sparse optical filter structure where the order of the filter is dictated by the largest $\Delta L_n$ and most of the filter taps are zero except for the present $\Delta L_i$ paths.

Here, we present a high-resolution wavemeter using a single-stream asymmetric MZI circuit with four different $\Delta L$ paths of increasing value (Fig. 3a). The minimum phase condition in this circuit is met by the use of a reference path that goes through a single-splitter operation instead of the two-splitter operations that the optical delay lines undergo. As a result, the power entering the reference path is guaranteed to exceed that entering the optical delay lines, ensuring the circuit meets the minimum phase condition.

As shown in Algorithm 2, the wavelength regression algorithm used for this design, follows the same procedure as the single $\Delta L$

design with a couple of adjustments needed to accommodate the extraction of the four different delay paths present in this structure. The extraction of $G(\omega)$ is carried out in the same way, while the calibration and wavelength models need to be adjusted. We note that we can perform the same mathematical treatment that was applied to the single $\Delta L$ design, where we can factor out the $e^{j\beta \Delta L1}$ term that provides a $G(\omega)$ comprised of four different $e^{j\beta \Delta Ln}$ terms and a constant term. Similarly, we filter out the DC term by applying the same $w_{min}(n)$ window function to extract $G(\omega)$.

The calibration process begins with extracting $G(\omega)$ for the multi-path MZI structure. The extraction of each delay line occurs by zero-padding the resulting $G(\omega)$ to allow the sinc-interpolated Fourier space enough resolution between $\Delta L1$ and $\Delta L2$ since $\Delta L2/\Delta L1 \approx 2$. We note that this optical path length difference is not sufficient to allow us to properly discern between these two optical paths in the Fourier space without sufficient zero-padding. In addition, $G_{copies}$ is the result of duplicating the $G(\omega)$ vector to buffer the filtering process ($h_{filter}$) similar to the single $\Delta L$ scenario. Once we have the location of the index of the four peaks, we initialize $\Theta$, $C$ to empty sets to allocate space for our filtered phases and model variables, respectively. Next, we iterate over the found peaks, and for each peak, we follow a similar filtering process to that presented for the single $\Delta L$ design. We define a filter centered on the found $i$-th peak and apply the filter to $G_{copies}$. Finally, the calibration process ends with modeling the unwrapped $i$-th filtered phase using a second-degree polynomial. Details regarding the need for a second-degree polynomial are shown in Supplementary Section 4.

The wavelength model for the multi-$\Delta L$ design requires the addition of the MZI order as we sweep from the smallest $\Delta L$ path (largest FSR) to the largest path (smallest FSR). To accomplish this, we follow a similar procedure as shown in ref. [13], where the initial MZI ($i = 0$) has an unambiguous phase, as it is uniquely determined in the bandwidth of interest. As a result, the calculation of frequency/wavelength deviation ($\Delta f$) is also unambiguous. However, as we iterate over the subsequent delay paths ($i > 0$), we need to find the delay path order ($\Delta m$) referenced by the previous delay path frequency deviation approximation. To do this, we first calculate the FSR of the current MZI using the previously derived phase model and the previous frequency deviation approximation, noting that

$$FSR = \frac{2\pi}{L}\left|\left(\frac{d\beta}{d\omega}\right)^{-1}\right| \approx \frac{2\pi}{C_2 \Delta f + C_1} \tag{6}$$

where $C_2$ and $C_1$ are the model coefficients found by the unwrapped phase model regression, and the $\beta$ expression is given by Eq. (5). Next, we find the relative frequency calculation ($\delta f$) using the wrapped filtered phase found in $\Theta[i]$. Finally, the order is found by comparing the current $i$-th filtered phase with the frequency deviation approximation found by the previous delay path normalized by the current $i$-th FSR. Subsequently, the absolute frequency deviation of the current $i$-th path is calculated by unwrapping the relative frequency using the calculated delay path order and FSR.

Figure 3b shows the photodetected signal, the retrieved phase using the Hilbert transform method, and the Fourier transform of the extracted $G(\omega)$ with the four delay lines being visible in the spectrum. The real-valued filter function is then constructed from the spectrum information, thereby providing the complex-valued taps that are used for the filtering operation. Figure 3c shows the resulting wrapped phases from each filtering operation, indicating a successful retrieval of the respective FSR from each delay line, such as Filter 1: 10 nm, Filter 2: 5 *nm*, Filter 3: 1 nm, and Filter 4: 0.1 nm. This phase is then unwrapped and modeled by a second-degree polynomial, which provides an RMSE of <0.06 *rad*. The wrapped phase is also used for the wavelength model described in the algorithm section. We note that the wavelength regression error is minimized as the algorithm goes through each filter stage with an RMSE of <1 pm for the final 0.1 pm stage. We also note

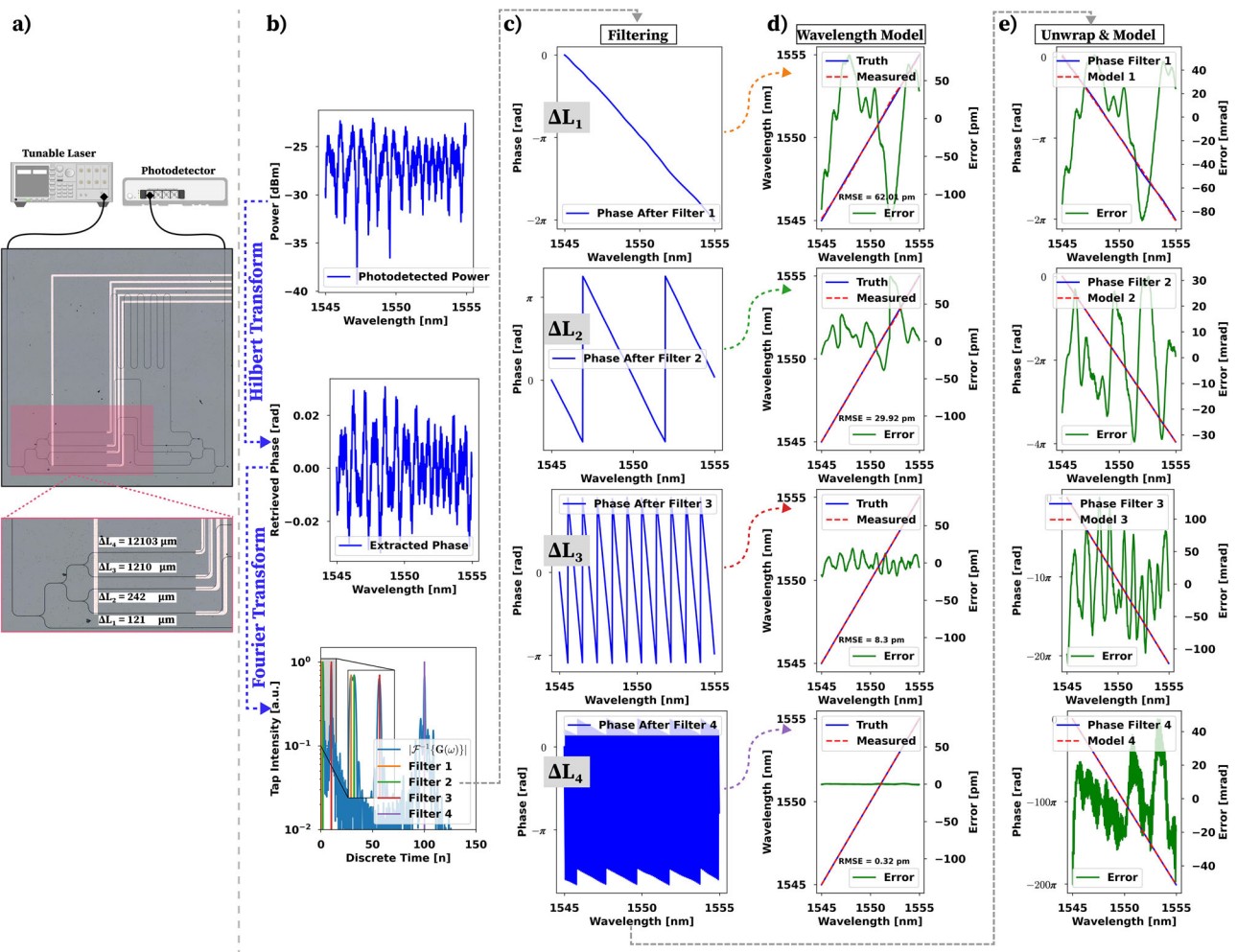

**Fig. 3 | Sub-picometer wavelength regression over arbitrary bandwidths using sparse filtering. a** Micrograph of the 4 delay path MZI circuit showing the experimental setup, including a Keysight 81606A tunable laser, and an N7744a photodetector. **b** Photodetected power, phase retrieval process through the Hilbert transform, and Fourier Transform of the complex-valued spectral response of the four delay paths showing the designed filter taps. **c** The filtered phase after each filter stage shows the uniquely determined phase ($\Delta L1$), and the subsequent delay stages used to increase the final measurement resolution. **d** Second-degree wavelength model application on the wrapped phases using the order of $\Delta L1$ to determine the orders of the subsequent $\Delta L$ paths showing an RMSE of < 1 pm in the final stage. **e** Unwrapped phases second-degree models showing an error of <0.06 rad.

that the FSR correction for the second (5 nm) and third (1 nm) stages causes small discontinuities at each phase wrapping point. This is because the second-degree polynomial used to calculate the wavelength is only needed for the fourth stage, as dispersion is more prominent in this case. As a result, the corrections made for the second and third stage overcorrect for this measurement, thereby causing these discontinuities. However, given the small FSR of the final stage, we see a minimization of these discontinuities in this last stage calculation, as this is the stage where dispersion is the most prominent and the second-degree function is most needed.

## Discussion

We have introduced a methodology for wavelength regression that exploits phase retrieval in a *minimum-phase* PIC architecture. By deriving and experimentally validating the general minimum-phase condition for asymmetric Mach-Zehnder interferometers (MZIs), we demonstrated that the retrieved phase agrees closely with the simulated phase and the analytically predicted minimum-phase boundary. Using four minimum-phase MZI designs, we implemented a phase-driven wavelength regression algorithm that achieved errors below 10 pm. The role of phase as the primary information carrier was confirmed experimentally: mismatches between training and testing

circuits produced larger errors due to phase slope differences between the calibrated and testing points.

A key limitation of the single–$\Delta L$ design is its sensitivity to global phase changes, restricting operation to a fixed bandwidth. To overcome this, we proposed and realized a multi–$\Delta L$ minimum-phase MZI design in which a short-delay path defines the operational bandwidth and longer-delay paths increase resolution. This architecture achieved <1 pm RMSE while maintaining the SNR advantage of a single-stream system (see Section 6 of the Supplement for a detailed system-level SNR analysis). We detailed the reconstruction algorithm, accounting for dispersion effects introduced by the longest delay path, and showed that a second-order phase model is necessary to minimize model error. This model also predicts the operational limits of the input and output waveguide couplers.

The current demonstration is limited to single-wavelength laser operation and does not yet include broadband sources. Moreover, Hilbert-transform phase retrieval requires a signal length greater than one, meaning that continuous single-wavelength operation in a minimum-phase framework remains an open research direction.

Additionally, the approach presented here utilizes a Fourier transformation to determine the delay of each path in the minimum-phase MZI circuit structure. The bandwidth of this minimum-phase

approach is constrained by the directional couplers' bandwidth, as changes in the coupling coefficient cause changes in the minimum-phase condition, which affect the phase-retrieval process. Because of interference at the MZI output coupler, the delay paths need to be carefully designed to avoid as much spectral leakage as possible in the Fourier domain. The ratio between any two subsequent delay paths must be designed so that the separation between the delay lines in the Fourier space can be accurately resolved within the bandwidth of interest. Furthermore, spectral leakage minimization is highly desired to avoid erroneous delay line extraction. To accomplish this, the increment between any two given subsequent delay paths must be an integer, such as $\Delta L_N/\Delta L_{N-1} = m$, where $m$ is an integer. Larger $m$ values provide better isolation between the delay lines in the Fourier space, thereby making the design more robust to spectral leakage. Further evidence of this process is shown in the Supplementary Section 3A, B.

Beyond wavelength metrology, this minimum-phase approach points toward *a broader design paradigm for sparse, phase-stable optical filters* and high-precision device characterization in integrated photonics. By combining phase retrieval, generalized minimum-phase conditions, and multi-delay architectures, the method opens a path to scalable, sub-picometer-resolution spectrometry without the SNR penalties of conventional multi-path systems, offering fertile ground for innovations in on-chip sensing, spectroscopy, and coherent optical signal processing.

## Methods

### Single $\Delta L$ experiment

We used AIM Photonics[43] to fabricate and package the designs shown in Fig. 2a. We designed 4 MZIs with different input coupling coefficients and $\Delta L$ to experimentally test the minimum-phase boundary, corresponding to the circuits marked by a red "x" in Fig. 1c. For the integrated directional couplers, we used the AIM Photonics PDK components for both the 50:50 splitters and the 90:10 splitters. Achieving larger bandwidths can be accomplished by designing couplers with the intended operational bandwidth using alternatives such as MMI or adiabatic couplers[44]. A Keithley 2400 SMU is used to measure the photodetector current at the output of each $K$ circuit at each wavelength using a Keysight 81606A tunable laser source (S1). The laser is swept from 1525 to 1530 nm in 1 pm increments, and the SMU is then used to measure the current at each wavelength point. We used a 128-point buffered measurement reporting the average. An uncalibrated semiconductor tunable laser diode (S2) and a Keysight DSOX1204g oscilloscope are used as the test laser source, as shown in Fig. 2a. The laser is set to continuously sweep the wavelength from 1525 to 1527.5 nm in 0.1 s. We put the scope capture window to 0.5 s, allowing us to capture multiple laser sweeps. We then note that the process used to retrieve the phase can also be used to discriminate the entire captured window, allowing us to identify the points where the data is valid. That is, the phase will be discontinuous where the wavelength jumps from 1527.5 to 1525 nm, so we use this discontinuity to limit the valid data window. On-chip loop-backs were used to measure coupling loss of around -7 *dB* per facet (which can be reduced by optimizing the fiber packaging process), and the propagation loss is estimated to be -1 dB/cm.

### Sparse filter experiment

We use the Applied Nanotools SiN MPW process to design and fabricate the multi-$\Delta L$ MZI circuit shown in Fig. 3. In this case, we choose SiN as our material platform owing to its small thermo-optic coefficient when compared to Si. The latter allows us to be less cognizant of temperature variations while still enabling the study of the minimum-phase approach without confounding factors such as mixed group indices designs[13]. For the integrated directional couplers, we use the 50:50 splitter within the ANT PDK. Similar to the single $\Delta L$ case, we could design MMI or adiabatic couplers that maintain splitting ratios over a larger bandwidth than conventional directional couplers[44]. The chosen $\Delta L$ values are 121 μm, 242 μm, 1210 μm, and 12103 μm

corresponding to FSR values of 10 nm, 5 nm, 1 nm, and 0.1 nm, respectively. Another MZI circuit was also designed with the following $\Delta L$ values: 12.1 μm, 15.12 μm, 60.51 μm, 121.03 μm, 302.58 μm, 605.16 μm, 1210.32 μm, 12103.27 μm with corresponding FSR values of 100 nm, 80 nm, 20 nm, 10 nm, 4 nm, 2 nm, 1 nm, 0.1 nm. We use an input and output coupler coefficient of 0.5 for both MZI structures. This is deemed sufficient for the minimum-phase condition, as we further split the delay lines 1:4, making the reference path have at least four times as much power as each delay line while also being the shortest. As a result, this is enough to meet the minimum-phase condition, as shown by both the simulation and the experimental results in the single $\Delta L$ case. We utilize edge coupling and a Maple Leaf automated test setup with a polarization-maintaining fiber array in combination with a Keysight 81606A tunable laser source and a Keysight N7744A photodetector. The coupling loss was measured at -9 dB per facet (which can be reduced with optimized edge coupler design), while the propagation loss was measured at -1 dB/cm. The laser source was swept from 1500 nm to 1600 nm in 1 pm steps. However, the data presented here are limited to the four delay lines circuit and the 10 nm window between 1545 nm and 1555 nm. Although the tunable laser supports sweeping from 1500 to 1600 nm, the demonstrated 10 nm operational bandwidth of the photonic IC is set by intentional delay-line design choices and by practical limitations of the directional couplers, rather than by the laser source itself. See Supplementary Section 3A, B for more discussion regarding this chosen circuit, wavelength window, and how to manage dispersion in the Fourier space.

## Data availability

The data generated in this study have been deposited in the following data repository: https://doi.org/10.6084/m9.figshare.30842513.

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

## Acknowledgements

The authors would like to thank Mark Indovina for insightful discussions on digital signal processing. S.P. and H.A.R.R. were supported for this research by the US National Institute of Standards and Technology (NIST) Rapid Assistance for Coronavirus Economic Response (RACER) program, grant number 70NANB22H015, as funded under the American Rescue Plan. S.P., H.A.R.R., and V.D. also acknowledge support in part by the Air Force Research Laboratory (FA8750-21-2-0004 and FA8750-25-2-0004) for this work. The views/conclusions contained herein are those of the authors and should not be interpreted as necessarily representing the official policies or endorsements, either expressed or implied, of the United States Air Force, AFRL, or the U.S. Government. Approved for public release (Distribution A: AFRL-2025-4263).

## Author contributions

H.A.R.R. conceived the study, developed the model, performed the simulations, and carried out the experimental procedures. L.N. contributed to data curation, manuscript writing, and discussion of the results. V.D. provided resources, including packaged chips and wirebonding support. S.P. supervised the research, conceived the experiments, contributed to result discussions, and assisted in manuscript preparation.

## Competing interests

The authors declare no competing interests.
