## [Transparent Peer Review file · Nature Communications]

Achieving Sub-pm Wavelength Regression via Minimum-Phase in a Single-Stream Photonic IC

Corresponding Author: Mr Hector Rubio

Version 0:

Reviewer comments:

Reviewer #1

(Remarks to the Author)

The paper introduces a single-stream, minimum-phase PIC architecture that uniquely reconstructs optical phase from intensity measurements via the Hilbert transform, eliminating the need for additional phase retrieval components. Additionally, the paper achieves sub-picometer root-mean-square wavelength regression error over arbitrary wavelength windows. The reviewer believes that several key issues remain that need to be clarified or justified, after which the manuscript can be considered for publication.

Q1: The phrase "Arbitrary Wavelength Windows" is used in the title of Part IV. While the multi- ΔL design does address wavelength ambiguity and allows operation over a much wider window than the FSR of a single large ΔL , the term "arbitrary" might be too absolute. The actual operating window is still limited by factors such as the bandwidth of the directional coupler and waveguide losses. It is suggested to modify the wording in the title or at the beginning of the section to be more precise.

Q2: The manuscript mentions "computational spectroscopy" in the abstract and introduction. However, all experiments in the paper are based on a single-wavelength, narrow-linewidth tunable laser. This demonstrates that the device can function as a high-precision wavemeter. Still, there is no evidence to support its capability in handling broadband sources, multi-line sources, or continuous spectrum analysis. It is recommended to clearly define the positioning as "high-precision wavelength metrology" and elaborate on the conditions and challenges of extending to complex spectra in the discussion.

Q3: Quantitative comparisons with conventional systems and recently proposed nanophotonics-enabled wavemeters are necessary.

Q4: The authors state in the introduction that conventional 90-degree hybrids "require a large footprint and are highly susceptible to manufacturing variations." However, the manuscript provides no evidence or analysis to substantiate the claim that the proposed method offers any advantages in either footprint or fabrication robustness.

Reviewer #2

(Remarks to the Author)

This paper introduces a single-stream, minimum-phase photonic integrated circuit (PIC) architecture, with core innovations including the minimum phase conditions and the Hilbert transform for phase retrieval. These contributions demonstrate substantial theoretical value through a systematic experimental design and offer implications for advancing next-generation computational spectroscopy technologies.

(1) Please clarify the novelty of the "generalized sparse tap configuration" compared to previous works. While the authors claim to generalize Xu et al.'s the minimum phase approach from sequential filter taps to arbitrary sparse taps, the distinction between the two could be made more explicit. Specifically, it is suggested to clearly explain how their design differs from existing works in terms of synthesis methods and functional advantages (including resolution, footprint, and signal/noise ratio). This will help to strengthen the manuscript's contributions.

(2) It is suggested to enhance the understandability of Hilbert transform-based phase retrieval. The key idea of the proposed research is the reliance on the minimum phase conditions and the Hilbert transform for phase retrieval, but the physical principles that enable unique optical phase reconstruction may be obscure.

(3) As shown in Fig. 2, the experimental results for wavelength regression prove that the experimental data drives the results and the filter is not biasing the proposed regression algorithm. Please explain in detail how the dataset improves the performance of the regression algorithm and how the dataset was constructed. How to improve both the measurement accuracy and the generalization of the proposed regression algorithm?

Reviewer #3

(Remarks to the Author)

Rubio, et al. have reported single input/output minimum phase PICs to extract optical phase from intensity measurement. They have presented the theory as well as some experimental results for single and multiple delay MZIs. Overall, the paper is well-written, and the idea is interesting, although the basic concept has been reported before. Below are my comments:

- Can the authors comment on using MMIs instead of directional couplers as they would make the system more wideband and less sensitive to fabrication variation (basically coupling ratio variations)? Especially for the sparse filter chip where 50/50 couplers are used.
- I suppose the chip is temperature stabilized. Is that the case? How would temperature change affect the measurement accuracy?
- There should be phase shifters within the MZIs to adjust/calibrate for phase mismatches and variations, but there is very little information on this. This should be explained in the paper, maybe in the methods.
- The PICs shown in the paper are fabricated in a SiN platform. Can the authors please comment on what the pros and cons of implementing the system on a silicon platform will be (in terms of loss, noise, etc.)?
- Some information about coupling and on-chip loss would be useful.
- The paragraph on page 4 explaining Fig. 2d and the figure itself are hard to follow. I suggest revising the text. Also, I think the correspondence between Fig 2c and 2d (arrows) is not very clear. I suggest revising that, too.
- I suggest changing the supplementary figure names to Fig. S1, S2, ..., or something similar, to avoid confusion with the main text. Also, it would be clearer if the lasers were referred to differently than the delay lengths in Fig. 1 (L1, L2).

Version 1:

Reviewer comments:

Reviewer #1

(Remarks to the Author)

I would like to thank the authors for their detailed responses and the significant revisions made to the manuscript. Most of my previous concerns regarding the methodology and the positioning of the work have been addressed. However, there are two remaining points regarding the scalability and the operational constraints of the proposed architecture that require further clarification before the manuscript is suitable for publication.

1. The authors demonstrate that resolution scales with the length and number of delay taps. However, the power of the reference arm must remain dominant to satisfy the stability of the Hilbert transform phase retrieval. As the number of delay taps increases, or as path lengths extend (leading to higher propagation losses), the power budget for each individual interferometric component is naturally squeezed. This inevitably constrains the system's dynamic range. It is recommended to discuss how the input power and the system's Signal-to-Noise Ratio impact the accuracy of the wavelength regression.

2. The manuscript notes that the tunable laser used in the experimental setup is capable of a wide sweep from 1500 nm to 1600 nm. However, the characterized operational range of the Photonic IC is limited to a 10 nm window. It is currently unclear whether this limitation is a fundamental design choice or a practical constraint.

Reviewer #2

(Remarks to the Author)

The authors have provided a comprehensive response to all comments and addressed all technical concerns. Their revisions enhance the manuscript's quality, resolving key issues and strengthening their contribution to the field of photonic integrated circuits (PICs).

1. The authors clearly distinguished their approach from the prior work proposed by Xu et al., demonstrating that a sequential tap design would require 100 delay lines for the same functionality as in the manuscript.
2. The manuscript introduces the work of Mecozzi et al., which provides the basic theory to link the minimum phase with the phase recovery from intensity measurements.
3. The manuscript has been revised to optimize the proposed wavelength regression algorithm in Figure 2 and enhance readability.

Reviewer #3

(Remarks to the Author)

I would like to thank the authors for addressing my comments. I believe the manuscript is stronger now and I do not have any further comments except for the following:

On page 8 (Methods) there is inconsistency in the loss per facet (7 dB vs 9 dB). Please fix that.

Version 2:

Reviewer comments:

Reviewer #1

(Remarks to the Author)

I thank the authors for their comprehensive response to all my comments and addressed all the technical concerns. I think the paper can be accepted for publication now.

Response to Reviewers:
Achieving Sub-pm Wavelength Regression via Minimum-Phase in a
Single-Stream Photonic IC

Hector A. Rubio Rivera^{1,*}, Lilian Neim¹, Venkatesh Deendayalan¹, Stefan Preble¹

¹Electrical and Microelectronic Engineering, Rochester Institute of Technology, NY, USA

*Corresponding author: hr8392@rit.edu

Reviewers' Comments and Author's Response

Dear Editor and Reviewers,

We thank you for the constructive feedback and the time you dedicated to reviewing our manuscript. In the peer review response document, we provide a detailed point-by-point response to all of the comments. As you will see, we have addressed each concern thoroughly and implemented extensive revisions to the manuscript, which we summarize here:

- **Reviewer 1. Comment 1.** Revised Section II.C title from “Sparse Filter Structure for Arbitrary Wavelength Windows” to “Sparse Filter Structure for Custom Wavelength Windows” to more accurately reflect realistic design constraints.
- **Reviewer 1. Comment 2.** Added a new Supplementary Section IV titled “Potential Application: Complex Fourier Transform Spectroscopy”, which demonstrates how the experimental calibration data from the main text can be used to reconstruct synthetic complex spectra. This illustrates the potential extension of the method to broadband and multi-line sources.
- **Reviewer 1. Comment 3.** Expanded the quantitative comparison to include conventional and recently reported integrated wavemeters (adapted from Yao *et al.*, *Nat. Commun.* 2023). The new figure highlights the bandwidth–resolution trade-offs and shows that the proposed single-stream minimum-phase approach achieves sub-picometer resolution without requiring multiple detection channels.
- **Reviewer 1. Comment 4.** Added new simulations of conventional 90° hybrid architectures to assess sensitivity to phase imbalance. The results confirm that small fabrication deviations can severely degrade hybrid-based approaches. The proposed minimum-phase method eliminates the need for phase-sensitive hybrids, offering improved fabrication robustness and reduced footprint.

- **Reviewer 2. Comment 1.** Clarified the novelty of the *generalized sparse tap configuration* compared to prior sequential designs (Xu et al, *Nat. Phot.* 2022). We added an explicit explanation in the Introduction and Supplemental Section III, showing that sequential taps would require ~ 100 delay lines to achieve the same resolution, whereas our sparse configuration reduces footprint and complexity while maintaining sub-pm resolution. The revised text emphasizes functional advantages in resolution, and footprint.
- **Reviewer 2. Comment 2.** Enhanced the description of Hilbert-transform-based phase retrieval by adding a clear explanation of the physical principles and citing Mecozzi’s work on necessary and sufficient conditions for minimum-phase systems. The Introduction now explicitly states that phase recovery is guaranteed when zeros and poles lie inside the unit circle, enabling unique phase reconstruction from intensity measurements.
- **Reviewer 2. Comment 3.** Revised Figure 2 for clarity and added a new Supplemental Section titled “II. Single ΔL : Calibration & Testing Algorithm”. This section details dataset construction, calibration workflow, and testing across 49 circuits from two packaged chips, demonstrating robustness to manufacturing variability. Subsection II.B in the main text was rewritten to align with the updated figure.
- **Reviewer 2. Comment 4.** Expanded discussion on how the dataset informs algorithm performance. Clarified that the dataset does not bias regression but enables visualization of calibration/testing combinations and RMSE trends, improving interpretability and validating generalization across chips.
- **Reviewer 3. Comment 1.** Added discussion on using MMIs or adiabatic couplers to improve bandwidth and reduce sensitivity to fabrication variations. Methods section now includes sentences highlighting these alternatives for future designs.
- **Reviewer 3. Comment 2.** Addressed temperature stabilization and its impact on accuracy. Added explanation in Subsection II.B and Methods section noting that global phase shifts manifest as small RMSE increases and can be corrected in post-processing. Discussed potential for athermal designs using complementary materials (Si/SiN).
- **Reviewer 3. Comment 3.** Clarified the role of on-chip phase shifters. While not required for wavemeter operation, they enable advanced applications such as complex Fourier-transform spectroscopy (demonstrated in Supplemental Section V). Added explanatory sentences in Methods.
- **Reviewer 3. Comment 4.** Compared Si vs. SiN platforms, outlining trade-offs in footprint, propagation loss, and thermal sensitivity. Added rationale for selecting SiN for this first demonstration in Methods section.

- **Reviewer 3. Comment 5.** Included details on coupling and propagation losses, supported by experimental measurements using the minimum-phase framework (see new Figure R4 in Supplemental). Added numerical values for insertion and propagation losses in Methods.
- **Reviewer 3. Comment 6.** Revised Figure 2 and corresponding text for clarity; added new supplemental section for calibration/testing workflow. Updated supplementary figure naming to Fig. S1, S2, etc., and renamed laser labels in Fig. 1 to avoid confusion with delay-line notation.

Overall, these changes have significantly enhanced the paper, and we hope you will agree that it is now suitable for publication in Nature Communications.

Reviewer 1

Remarks to the Author:

The paper introduces a single-stream, minimum-phase PIC architecture that uniquely reconstructs optical phase from intensity measurements via the Hilbert transform, eliminating the need for additional phase retrieval components. Additionally, the paper achieves sub-picometer root-mean-square wavelength regression error over arbitrary wavelength windows. The reviewer believes that several key issues remain that need to be clarified or justified, after which the manuscript can be considered for publication.

Response:

We thank the reviewer for their supportive feedback and insightful suggestions. We have addressed each of their points below.

Reviewer Comment 1:

The phrase "Arbitrary Wavelength Windows" is used in the title of Part IV. While the multi- ΔL design does address wavelength ambiguity and allows operation over a much wider window than the FSR of a single large ΔL , the term "arbitrary" might be too absolute. The actual operating window is still limited by factors such as the bandwidth of the directional coupler and waveguide losses. It is suggested to modify the wording in the title or at the beginning of the section to be more precise.

Response:

We thank Reviewer 1 for their suggestion, and we agree that the term "arbitrary" may be too absolute, as it suggests that the methodology applies to any wavelength window, regardless of component design or limitations. To clarify this, we have modified the wording in Section II.C. Specifically, the title has been changed from "Sparse Filter Structure for Arbitrary Wavelength Windows" to "Sparse Filter Structure for Custom Wavelength Windows", reflecting that the design methodology applies to wavelength ranges achievable given the properties of the implemented circuit elements.

Reviewer Comment 2:

The manuscript mentions "computational spectroscopy" in the abstract and introduction. However, all experiments in the paper are based on a single-wavelength, narrow-linewidth tunable laser. This demonstrates that the device can function as a high-precision wavemeter. Still, there is no evidence to support its capability in handling broadband sources, multi-line sources, or continuous spectrum analysis. It is recommended to clearly define the positioning

as "high-precision wavelength metrology" and elaborate on the conditions and challenges of extending to complex spectra in the discussion

Response:

We thank Reviewer 1 for their comment on the applicability of our work in the computational spectroscopy field. Since submitting the manuscript, we have worked to demonstrate the broader applicability of our approach to computational spectroscopy. Specifically, we have now applied the minimum-phase design framework to enable complex-valued Fourier-transform spectroscopy (FTS) for reconstructing various types of spectra in the same multi- ΔL single-stream circuit described in subsection II.C of the main article. By utilizing minimum-phase, we overcome the manufacturing sensitivity of conventional FTS, which has hindered reconstruction robustness [1]. Specifically, like in the wavemeter algorithm, the minimum-phase design framework enables phase retrieval directly from the intensity measurement [2]. In doing so, we have directly fit the spectral phase function, thereby absorbing manufacturing variations in our calibration procedure. Furthermore, owing to the complex-valued nature of our approach, we achieved lossless interpolation of the interferogram alignment by leveraging phase slope multiplication in the complex domain. Consequently, these features have enabled us to reconstruct both the amplitude and the phase of broadband sources.

We have highlighted this example of complex-valued reconstruction computational spectroscopy in a new section of the Supplemental Information (Section V. Application: Complex Fourier Transform Spectroscopy). It utilizes the experimental data from Subsection II.C of the main paper (Multi-Delta-L / Sparse Filter Structure for Custom Wavelength Windows) to showcase the reconstruction of complex spectra produced by a chirped Gaussian, Mach-Zehnder Interferometer, or Ring Resonator. As shown in Fig. R1(a) (Fig. S6 in the supplemental), the experimental configuration closely mirrors that of Figure 3 in the main article. In order to realize the complex-valued FTS functionality, thermo-optic tuning was performed using DC probes and a Keithley 2400 SMU supplying 0–10 V in 0.5 V steps, with a wavelength sweep conducted at each voltage level.

Using the circuit's minimum-phase property, the phase response is derived from the measured intensity through a Hilbert transform (Fig. R1(b,c)). Digital filtering around each delay line then isolates the individual ΔL interferometers (Fig. R1(d)). The filtered outputs reveal the spectral phase response as a function of the electrical power applied to each phase shifter (Fig. R1(e–h)). This experimental phase information allows direct characterization of the optical path delay sweep, $\Delta L(P)$, which is subsequently modeled using a polynomial fit (Fig. R1). Based on this delay model, the interferograms are aligned by applying the phase response expressed as $\propto e^{\beta \Delta L_i(P)}$ (Fig. R1(m–p)). Finally, the Fourier transform of the averaged aligned interferograms produces the reconstructed

Figure R1: **Spectrometer Experimental Calibration.** (a) Experimental setup for gathering the spectrometer calibration data using a Keysight tunable laser and PD combo with a Keithley 2400 Source-Measure-Unit (SMU). (b) Photodetected power and (c) retrieved phase using the Hilbert transform and leveraging the minimum-phase design of the resulting MZI system. (d) The inverse Fourier Transform of the constructed complex-valued response allows us to filter each ΔL delay line for further post-processing. The application of each filter provides the phase of each ΔL_i (e-h). Subsequently, each spectral phase is fitted using a linear model to obtain $\Delta L(P)$ (i-l). The data from this is then used to align each interferogram by using the phase factor $\propto \Delta L(P)/c$ at each electrical power value (m-p).

spectrum. The reconstruction for the three cases (chirped Gaussian, Mach-Zehnder

Interferometer, Micro-ring resonator), along with the calibration step, is presented in Figure S6 of the Supplemental Section V. Application: Complex Fourier Transform Spectroscopy.

Reviewer Comment 3:

Quantitative comparisons with conventional systems and recently proposed nanophotonics-enabled wavemeters are necessary.

Response:

We performed a quantitative comparison with prior work. Integrated wavemeter demonstrations are rare, and since our method introduces a new approach for high-precision wavelength metrology with broad relevance to computational spectroscopy, we compared it against all conventional integrated spectrometer designs. Figure R2 adapts a recent comparison by Yao et al. [3] (with additional publications as recent as of this writing), highlighting different spectrometer types, including wavemeters, and showing that our method achieves a new resolution regime.

Direct comparisons to conventional computational spectroscopy can be misleading because of fundamental differences in methodology. Traditional techniques generally fall into three categories: narrowband filtering, Fourier Transform Spectroscopy (FTS), and reconstruction-based methods. These approaches segment the spectrum either spatially (narrowband filtering, reconstruction) or temporally (FTS), imposing design constraints such as multiple photodetectors or parallel measurement channels. For example, Yao et al. [3] (Fig. 5(b)) report footprint per channel, whereas our method performs regression on retrieved spectral phase, eliminating per-channel footprint constraints. The spectral phase serves as a linear basis for regression, and the sweep rate and resolution define the effective number of channels. Bandwidth and resolution remain useful metrics, and our approach notably surpasses the sub-pm resolution limit. Prior interferometric phase-based methods [4, 5] required multiple detection channels, while ours achieves single-stream operation with a higher bandwidth-to-resolution ratio.

Because these comparisons are somewhat arbitrary, we have excluded them from the revised manuscript and supplementary material. Interested readers can access this information in this peer review file once the manuscript is published.

Figure R2: **Bandwidth and Resolution Comparison with Relevant Work; Adapted from [3] Fig. 5(a)**. Comparison of our methodology (red star) with current state-of-the-art works in computational spectroscopy with approaches encompassing Fourier Transform Spectroscopy (FTS), spectrum reconstruction, and narrowband filtering (blue dots), along with phase-based wavelength metrology (green dots).

Reviewer Comment 4:

The authors state in the introduction that conventional 90-degree hybrids "require a large footprint and are highly susceptible to manufacturing variations." However, the manuscript provides no evidence or analysis to substantiate the claim that the proposed method offers any advantages in either footprint or fabrication robustness

Response:

We thank the reviewer for their thoughtful comment regarding the fundamental comparison between 90-degree hybrids and our approach. Here, we will demonstrate that 90-degree hybrids are sensitive to phase mismatches as small as 1.1° . This limitation narrows their applicability for use in wavelength metrology. As reported in [4], a phase imbalance exceeding 1.1° can significantly degrade the resolution of 90° hybrid-based approaches. To assess robustness, we simulated a conventional 90° hybrid architecture using Lumerical Interconnect [6] under ideal conditions. We modeled three MZIs on a SiN platform — analogous to our multi- ΔL design but employing 90° hybrids for phase extraction rather than minimum-phase reconstruction. We swept the hybrid phase imbalance from 88° to 92° . The results, shown in Fig. R3, highlight the sensitivity of this approach to small phase deviations. We also note that the oscillatory phase-retrieval behavior induced by phase imbalance can be observed in Fig. 8 of [4] and was subsequently identified as a reproducible error source in their Fig. 9.

Experimentally, prior work has demonstrated low-loss 90° hybrids [7]; however, their measured phase imbalances exceed the thresholds required for the high resolution reported in [4]. In contrast, our minimum-phase methodology retrieves the spectral phase directly from intensity measurements rather than relying on phase-sensitive I/Q outputs, providing intrinsic robustness to fabrication-induced phase errors. This robustness is further supported by Fig. 2 of the manuscript, which presents cross-chip calibration results between two independently packaged chips (P1, P2). All calibration combinations maintain accurate wavelength estimation, demonstrating that the proposed method tolerates manufacturing variation without the strict phase-matching requirements of 90° hybrids. As far as the footprint goes, 90° hybrids require, at a minimum, 4 couplers. In our approach, we don't need additional components to enable phase retrieval, as the minimum-phase approach allows us to obtain the phase from intensity measurements directly.

Figure R3: **90-degree Hybrid Phase Noise Robustness Simulation.**(a) Interconnect simulation setup using ideal components for the 90° hybrid component. The phase is calculated as $\theta = \tan^{-1}(Q/I)$. (b) simulation results showing three different MZI configurations corresponding to FSRs of 10 nm (121 μm), 5 nm (242 μm), and 1nm (1210 μm) with varying levels of phase imbalance (ϕ) to simulate manufacturing robustness under ideal conditions. The phase offset is calculated $\Delta\theta = \theta_i - \theta_{90}$ where θ_i is calculated at ϕ_i , and θ_{90} is at $\phi = 90^\circ$.

Reviewer 2

Remarks to the Author:

This paper introduces a single-stream, minimum-phase photonic integrated circuit (PIC) architecture, with core innovations including the minimum phase conditions and the Hilbert transform for phase retrieval. These contributions demonstrate substantial theoretical value through a systematic experimental design and offer implications for advancing next-generation computational spectroscopy technologies.

Response:

We thank the reviewer for their thoughtful evaluation of our work and for recognizing the theoretical significance of our design methodology. We appreciate the positive assessment of the core innovations presented in the manuscript. Below, we provide detailed responses to each of the reviewers' comments.

Reviewer Comment 1:

Please clarify the novelty of the “generalized sparse tap configuration” compared to previous works. While the authors claim to generalize Xu et al.’s the minimum phase approach from sequential filter taps to arbitrary sparse taps, the distinction between the two could be made more explicit. Specifically, it is suggested to clearly explain how their design differs from existing works in terms of synthesis methods and functional advantages (including resolution, footprint, and signal/noise ratio). This will help to strengthen the manuscript’s contributions.

Response:

We thank the reviewer for their comment regarding the novelty of our work. Here, we will demonstrate the distinction by reviewing Xu’s work and showing that, if we were to use sequential taps, it would require 100 delay lines to realize the same functionality as in our manuscript. Specifically, Xu et al. focused solely on using the minimum-phase approach with the intention of synthesizing FIR filters leveraging programmable photonics as their framework. This limits the scope of their demonstration to the case where we have an interference pattern that is composed of a sum given by

$$H(\omega) = \sum_n e^{in\omega}, \forall n \in [0, 1, 2, 3, \dots, N] \quad (1)$$

where ω is the radial frequency. We observe that this is a sum that converges to $H(\omega) = (1 - e^{iN\omega})/(1 - e^{i\omega})$. This fact makes the tap extraction process for their circuit a simple Fourier transformation. No preprocessing is needed, as the mathematical definition automatically accounts for the sequential delay increases. In contrast, in our work, the delay lines need to increase in a non-sequential manner. This is due to the

need to increase our delay lines to ensure a higher resolution wavelength measurement. As shown in the Supplemental in Section III, we had to condition our signal properly such that we could extract our filter taps. In addition, we also demonstrate, in the same Section, that the non-sequential filter taps need to follow a specific structure for proper signal reconstruction and modeling of the resulting $G(\omega)$. Specifically, Fig. S4 shows that the fabricated 8-tap filter does not fulfill the set of necessary conditions for proper signal reconstruction as opposed to our 4-tap filter design. This work is a fundamental generalization of Xu’s work, as this analysis was not provided in their work due to the nature of their demonstration. For our case, following their approach would have resulted in a circuit that requires at least 100 delay paths to achieve the same reported sub-pm resolution. To address all these points, we have modified the Introduction of the main paper. Specifically, we have added: “This conceptual limitation would prohibitively increase a spectrometer circuit’s footprint by n -fold (n delays paths needed to go from bandwidth-defining FSR to higher-resolution FSR). Here, we overcome this through the use of non-sequential tap increases, uncovering the necessary optical signal processing needed to achieve high-performance wavelength metrology. This work enables arbitrary and sparse tap filter configurations, where n can be any integer, removing the footprint requirements from n delay paths to any $m < n$. Enabling this arbitrary increase in delay length significantly enhances spectral resolution, proving particularly valuable for wavelength metrology applications.”

Reviewer Comment 2:

It is suggested to enhance the understandability of Hilbert transform-based phase retrieval. The key idea of the proposed research is the reliance on the minimum phase conditions and the Hilbert transform for phase retrieval, but the physical principles that enable unique optical phase reconstruction may be obscure.

Response:

We thank the reviewer for highlighting the need to clarify the physical principles underlying our Hilbert-transform-based phase retrieval approach. We acknowledge that the existing literature on this topic is relatively limited, which may contribute to the perceived obscurity of the concept. To address this, we have incorporated a reference to the work by Mecozzi[8], which rigorously derives the necessary and sufficient conditions for accurate phase reconstruction from intensity measurements. As discussed in our manuscript, these conditions ensure that when both the zeros and poles of the system lie inside the unit circle, the system is minimum phase, enabling phase recovery via the Hilbert transform. We have added this reference to the Introduction and revised the text to explicitly state: “Specifically, the phase is determined directly from the magnitude via their fundamental relationship in a minimum phase system, enforced by the Hilbert transform.[8, 9]”.

Reviewer Comment 3:

As shown in Fig. 2, the experimental results for wavelength regression prove that the experimental data drives the results and the filter is not biasing the proposed regression algorithm. Please explain in detail how the dataset improves the performance of the regression algorithm and how the dataset was constructed. How to improve both the measurement accuracy and the generalization of the proposed regression algorithm?

Response:

We thank the reviewer for their insightful comment regarding the clarification of our wavelength regression algorithm, as presented in Figure 2 of the main manuscript. To improve clarity, we have extensively revised Figure 2 to streamline the workflow and added a detailed description of the calibration and testing procedures in a new section of the Supplemental Material titled: “II. Single \$\Delta L\$: Calibration & Testing Algorithm”. Additionally, Subsection B of Section II (Results) has been rewritten to align with the updated figure. Regarding the reviewer’s specific question, the dataset is not intended to enhance measurement accuracy or generalization of the proposed algorithm. Instead, it serves as a structured representation of data collected from 49 circuits across two packaged chips (each containing four chips, with one circuit (K_4 in P_2) excluded due to a short circuit). The matrix format enables us to visualize calibration and testing across different circuit combinations (rows correspond to calibration circuits $P_i - K_j$, columns to testing circuits), allowing an assessment of chip-to-chip variation and, more importantly, the robustness of the algorithm to manufacturing variability.

Reviewer 3

Remarks to the Author:

Rubio, et al. have reported single input/output minimum phase PICs to extract optical phase from intensity measurement. They have presented the theory as well as some experimental results for single and multiple delay MZIs. Overall, the paper is well-written, and the idea is interesting, although the basic concept has been reported before. Below are my comments:

Response:

We appreciate the reviewer's time and effort in evaluating our work. We are grateful for the positive feedback regarding the clarity of our manuscript and the interest in our research. Below, we provide detailed responses to each comment in the following sections.

Reviewer Comment 1:

Can the authors comment on using MMIs instead of directional couplers as they would make the system more wideband and less sensitive to fabrication variation (basically coupling ratio variations)? Especially for the sparse filter chip where 50/50 couplers are used

Response:

We thank the reviewer for their comment regarding the design complexity that arises from bandwidth-limited devices such as directional couplers. We used PDK (Process Design Kit) devices for the couplers in both circuits in the manuscript, the single ΔL and the multi- ΔL architectures. Overall, broadband devices such as MMIs and adiabatic couplers[10] would allow us to maintain a larger bandwidth where minimum-phase is maintained, thereby enabling larger operating bandwidths for the wavemeter. We added the following sentence to Subsection A of the Methods section of the main paper: “For the integrated directional couplers, we used the AIM Photonics PDK components for both the 50:50 splitters and the 90:10 splitters. Achieving larger bandwidths can be accomplished by designing couplers with the intended operational bandwidth using alternatives such as MMI or adiabatic couplers”. For Subsection B of the Methods section, we added the following sentence: “For the integrated directional couplers, we use the 50:50 splitter within the ANT PDK. Similar to the single ΔL case, we could design MMI or adiabatic couplers that maintain splitting ratios over a larger bandwidth than conventional directional couplers”.

Reviewer Comment 2:

I suppose the chip is temperature stabilized. Is that the case? How would temperature change affect the measurement accuracy?

Response:

We thank the reviewer for their comments regarding temperature stabilization and the implications on measurement accuracy. We did use temperature stabilization for both chips. However, global phase shifts due to temperature variations become apparent in the results shown in Figure 2.(d) from the main paper. The slight RMSE increases found in the matrices when both the calibration and testing data match in ΔL , i.e. $K_1 = K_3$ and $K_2 = K_4$ are due to this temperature instability. We added the the following sentence in Subection II.B to point this out: “Further evidence of this effect is shown in Figure 2(c), where the matched ΔL testing data (red arrow) presents a small global phase offset when compared to the calibrated data (blue arrow), producing an RMSE increase”.

In the multi- ΔL design, the phase is uniquely determined over the bandwidth of interest, so temperature changes manifest only as a global phase offset that can be corrected in post-processing. A more robust approach would be to use materials with complementary thermo-optic coefficients, such as Si and SiN, to achieve true athermal operation [4]. However, implementing such a scheme could introduce additional complexity due to differing group indices, potentially confounding the demonstration of the minimum-phase approach. Since Stern et al. already demonstrated athermal operation [4], our focus here is on showcasing the capabilities of the minimum-phase methodology for high-precision wavelength metrology. We have addressed this point in conjunction with Comment 4 (below) by adding the following sentences to Subsection B in the Methods section of the main paper: “In this case, we choose SiN as our material platform owing to its low loss and small thermo-optic coefficient when compared to Si. The latter allows us to be less cognizant of temperature variations while still enabling the study of the minimum-phase approach without confounding factors such as mixed group indices designs.”

Reviewer Comment 3:

There should be phase shifters within the MZIs to adjust/calibrate for phase mismatches and variations, but there is very little information on this. This should be explained in the paper, maybe in the methods.

Response:

We thank the reviewer for their thoughtful consideration of the on-chip phase shifters for calibrating for phase mismatches. While this could be done, it actually isn't required

for wavemeter operation. Specifically, the minimum-phase reconstruction algorithm can identify any phase imbalances within the circuit. As an example of this, in Section IV of the supplemental, we demonstrated that the bandwidth of the PDK coupler could be extracted by monitoring the phase imbalance as a function of wavelength. Consequently, since the reconstruction algorithm can identify these phases, an explicit thermo-optic control isn't required. With that said, in our response to Reviewer 1 (Comment 2), we have now demonstrated that the on-chip phase shifters enable a new paradigm within Fourier Transform computational spectroscopy applications. Specifically, in the Supplemental, we have demonstrated how these phase shifters enable reconstruction of arbitrary complex spectra, such as a Chirped Gaussian, Ring Resonator, and Mach-Zehnder spectra. We plan to submit a detailed follow-up manuscript that will utilize this concept.

Reviewer Comment 4:

The PICs shown in the paper are fabricated in a SiN platform. Can the authors please comment on what the pros and cons of implementing the system on a silicon platform will be (in terms of loss, noise, etc.)?

Response:

We appreciate the reviewer's insightful question regarding the advantages and disadvantages of using the SiN/Si platforms. The experiments presented in the paper show both Si (single ΔL) and SiN (multi- ΔL) material platforms. Implementing the multi- ΔL on Si would generally allow for tighter integration and potentially smaller footprints due to the higher index contrast, but would come with trade-offs. In particular, Si typically exhibits higher propagation loss in the telecom band than SiN, and stronger thermo-optic effects, which could increase sensitivity to temperature variations. SiN, on the other hand, offers low propagation loss, wide transparency, and reduced thermal sensitivity, which are advantageous for high-precision wavelength metrology. Overall, the choice of platform involves balancing footprint, loss, thermal stability, and fabrication maturity. Here, SiN was selected to optimize for low loss and stability because this is the first demonstration of this methodology. We added the following sentence in Subsection B within the Methods section of the main paper: "In this case, we choose SiN as our material platform owing to its small thermo-optic coefficient when compared to Si. The latter allows us to be less cognizant of temperature variations while still enabling the study of the minimum-phase approach without confounding factors such as mixed group indices designs."

Reviewer Comment 5:

Some information about coupling and on-chip loss would be useful.

Response:

We thank the reviewer for this suggestion. The on-chip devices exhibit typical propagation losses for the SiN platform, with waveguide losses of ~ 1 dB/cm. Fiber-to-chip coupling was achieved using edge couplers with insertion losses of 9dB per facet (which can be improved upon with optimization to the fiber packaging process). We have added this information to the Methods section to provide a clearer picture of the system performance. To confirm the waveguide loss, we actually investigated and found that our minimum-phase approach allowed us to obtain the SiN platform loss by measuring the intensity of the four taps in the discrete-time domain and fitting the loss as a function of length, as shown in Figure R4. As a result of the reviewer’s thoughtful suggestion, we added the following sentences regarding loss in Subsection A within the Methods section of the main paper: On-chip loopbacks were used to measure coupling loss of around \$\sim 7\$ [dB] per facet (which can be reduced by optimizing the fiber packaging process), and the propagation loss is estimated to be \$\sim 1\$ [dB/cm], and in Subsection B within the Methods section of the main paper: The coupling loss was measured at \$\sim 9\$ [dB] per facet (which can be reduced with optimized edge coupler design), while the propagation loss was measured at \$\sim 1\$ [dB/cm]

Figure R4: **SiN Loss Measurement Using Minimum-Phase Framework.**(a) Temporal response of 4-tap SiN chip after phase retrieval. (b) linear fit of the normalized tap energy vs tap length, providing the chip loss per unit length.

Reviewer Comment 6:

The paragraph on page 4 explaining Fig. 2d and the figure itself are hard to follow. I suggest revising the text. Also, I think the correspondence between Fig. 2c and 2d (arrows) is not very clear. I suggest revising that, too.

Response:

We thank the reviewer for pointing out that the explanation of Fig. 2d and its correspondence with Fig. 2c could be clearer. We have modified Figure 2 entirely to streamline and simplify the presentation of our algorithm. In addition, we added a new section in the supplemental titled “II. Single ΔL : Calibration & Testing Algorithm” (See Reviewer 2, Comment 3). In addition we have revised Subsection B in Section II (Results) of the main paper to reflect the streamlined Figure 2. These changes aim to make the workflow and correspondence between the two panels more intuitive.

Reviewer Comment 7:

I suggest changing the supplementary figure names to Fig. S1, S2, . . . , or something similar, to avoid confusion with the main text. Also, it would be clearer if the lasers were referred to differently than the delay lengths in Fig. 1 (L1, L2).

Response:

We thank the reviewer for the helpful suggestion. We have updated the supplementary figure labels to Fig. S1, Fig. S2, etc., to distinguish them from figures in the main text. In addition, we renamed the laser labels in Fig. 1 from (L1, L2) to (S1, S2) to avoid confusion with the delay-line notation (L1, L2), and we revised the caption accordingly for clarity.

References

- [1] Li, A., Davis, J., Grieco, A., Alshamrani, N. & Fainman, Y. Fabrication-tolerant Fourier transform spectrometer on silicon with broad bandwidth and high resolution. Photonics Research **8**, 219–224 (2020). URL <https://opg.optica.org/prj/abstract.cfm?uri=prj-8-2-219>. Publisher: Optica Publishing Group.
- [2] Xu, X. et al. Self-calibrating programmable photonic integrated circuits. Nature Photonics **16**, 595–602 (2022). URL <https://www.nature.com/articles/s41566-022-01020-z>.
- [3] Yao, C. et al. Integrated reconstructive spectrometer with programmable photonic circuits. Nature Communications **14**, 6376 (2023). URL <https://www.nature.com/articles/s41467-023-42197-3>.

- [4] Stern, B., Kim, K., Gariah, H. & Bitauld, D. Athermal silicon photonic wavemeter for broadband and high-accuracy wavelength measurements. *Optics Express* **29**, 29946–29959 (2021). URL <https://opg.optica.org/oe/abstract.cfm?uri=oe-29-19-29946>.
- [5] Coggrave, C. R. et al. Single-shot wavelength meter on a chip based on exponentially increasing delays and in-phase quadrature detection. *Optics and Lasers in Engineering* **178**, 108163 (2024). URL <https://www.sciencedirect.com/science/article/pii/S0143816624001428>.
- [6] Inc., L. Lumerical interconnect: Photonic integrated circuit simulator.
- [7] Guan, H. et al. Compact and low loss 90° optical hybrid on a silicon-on-insulator platform. *Opt. Express* **25**, 28957–28968 (2017). URL <https://opg.optica.org/oe/abstract.cfm?URI=oe-25-23-28957>.
- [8] Mecozzi, A. A necessary and sufficient condition for minimum phase and implications for phase retrieval (2016). URL <http://arxiv.org/abs/1606.04861>. ArXiv:1606.04861 [cs].
- [9] Mecozzi, A., Antonelli, C. & Shtaif, M. Kramers–kronig coherent receiver. *Optica* **3**, 1220–1227 (2016). URL <https://opg.optica.org/optica/abstract.cfm?URI=optica-3-11-1220>.
- [10] Cabanillas, F. & Maria, J. Rapid adiabatic devices enabling integrated electronic-photonic quantum systems on chip (2022). URL <https://hdl.handle.net/2144/44749>.
- [11] Du, J. et al. High-resolution on-chip fourier transform spectrometer based on cascaded optical switches. *Opt. Lett.* **47**, 218–221 (2022). URL <https://opg.optica.org/ol/abstract.cfm?URI=ol-47-2-218>.
- [12] Emadi, A., Wu, H., de Graaf, G. & Wolffenbuttel, R. Design and implementation of a sub-nm resolution microspectrometer based on a linear-variable optical filter. *Opt. Express* **20**, 489–507 (2012). URL <https://opg.optica.org/oe/abstract.cfm?URI=oe-20-1-489>.
- [13] He, X. et al. Fully packaged on-chip ring resonator spectrometer. *Opt. Express* **33**, 30594–30606 (2025). URL <https://opg.optica.org/oe/abstract.cfm?URI=oe-33-14-30594>.
- [14] Kyotoku, B. B. C., Chen, L. & Lipson, M. Sub-nm resolution cavity enhanced microspectrometer. *Opt. Express* **18**, 102–107 (2010). URL <https://opg.optica.org/oe/abstract.cfm?URI=oe-18-1-102>.
- [15] Ma, X., Li, M. & He, J.-J. Cmos-compatible integrated spectrometer based on echelle diffraction grating and msm photodetector array. *IEEE Photonics Journal* **5**, 6600807–6600807 (2013).
- [16] Velasco, A. V. et al. High-resolution fourier-transform spectrometer chip with microphotonic silicon spiral waveguides. *Opt. Lett.* **38**, 706–708 (2013). URL <https://opg.optica.org/ol/abstract.cfm?URI=ol-38-5-706>.

- [17] Xia, Z. *et al.* High resolution on-chip spectroscopy based on miniaturized microdonut resonators. Opt. Express **19**, 12356–12364 (2011). URL <https://opg.optica.org/oe/abstract.cfm?URI=oe-19-13-12356>.
- [18] Yi, D., Zhang, Y., Wu, X. & Tsang, H. K. Integrated multimode waveguide with photonic lantern for speckle spectroscopy. IEEE Journal of Quantum Electronics **57**, 1–8 (2021).
- [19] Zheng, S. *et al.* A single-chip integrated spectrometer via tunable microring resonator array. IEEE Photonics Journal **11**, 1–9 (2019).
- [20] Bao, J. & Bawendi, M. G. A colloidal quantum dot spectrometer. Nature **523**, 67–70 (2015). URL <https://www.nature.com/articles/nature14576>. Publisher: Nature Publishing Group.
- [21] Chen, C., Gu, H. & Liu, S. Ultra-simplified diffraction-based computational spectrometer. Light: Science & Applications **13**, 9 (2024). URL <https://www.nature.com/articles/s41377-023-01355-4>.
- [22] Finco, G. *et al.* Monolithic thin-film lithium niobate broadband spectrometer with one nanometre resolution. Nature Communications **15**, 2330 (2024). URL <https://www.nature.com/articles/s41467-024-46512-4>. Publisher: Nature Publishing Group.
- [23] Hadibrata, W., Noh, H., Wei, H., Krishnaswamy, S. & Aydin, K. Compact, high-resolution inverse-designed on-chip spectrometer based on tailored disorder modes. Laser & Photonics Reviews **15**, 2000556 (2021). URL <https://onlinelibrary.wiley.com/doi/abs/10.1002/lpor.202000556>.
- [24] Kita, D. M. *et al.* High-performance and scalable on-chip digital Fourier transform spectroscopy. Nature Communications **9**, 4405 (2018). URL <https://doi.org/10.1038/s41467-018-06773-2>.
- [25] Li, A. *et al.* An integrated single-shot spectrometer with large bandwidth-resolution ratio and wide operation temperature range. Photonix **4**, 29 (2023). URL <https://doi.org/10.1186/s43074-023-00109-0>.
- [26] Li, A. & Fainman, Y. On-chip spectrometers using stratified waveguide filters. Nature Communications **12**, 2704 (2021). URL <https://www.nature.com/articles/s41467-021-23001-6>.
- [27] Lin, Z. *et al.* High-performance, intelligent, on-chip speckle spectrometer using 2d silicon photonic disordered microring lattice. Optica **10**, 497–504 (2023). URL <https://opg.optica.org/optica/abstract.cfm?uri=optica-10-4-497>.
- [28] Piels, M. & Zibar, D. Compact silicon multimode waveguide spectrometer with enhanced bandwidth. Scientific Reports **7**, 43454 (2017). URL <https://www.nature.com/articles/srep43454>. Publisher: Nature Publishing Group.

- [29] Redding, B., Liew, S. F., Sarma, R. & Cao, H. Compact spectrometer based on a disordered photonic chip. Nature Photonics **7**, 746–751 (2013). URL <https://www.nature.com/articles/nphoton.2013.190>. Publisher: Nature Publishing Group.
- [30] Redding, B., Liew, S. F., Bromberg, Y., Sarma, R. & Cao, H. Evanescently coupled multimode spiral spectrometer. Optica **3**, 956–962 (2016). URL <https://opg.optica.org/optica/abstract.cfm?uri=optica-3-9-956>.
- [31] Ruocco, A. & Bogaerts, W. Fully integrated soi wavelength meter based on phase shift technique. In 2015 IEEE 12th International Conference on Group IV Photonics (GFP), 131–132 (2015). URL <https://ieeexplore.ieee.org/document/7305985>.
- [32] Souza, M. C. M. M., Grieco, A., Frateschi, N. C. & Fainman, Y. Fourier transform spectrometer on silicon with thermo-optic non-linearity and dispersion correction. Nature Communications **9**, 665 (2018). URL <https://www.nature.com/articles/s41467-018-03004-6>. Publisher: Nature Publishing Group.
- [33] Wang, H., Lin, Z., Li, Q. & Shi, W. On-chip fourier transform spectrometers by dual-polarized detection. Optics Letters **44**, 2923–2926 (2019). URL <https://opg.optica.org/ol/abstract.cfm?uri=ol-44-11-2923>.
- [34] Xu, H., Qin, Y., Hu, G. & Tsang, H. K. Breaking the resolution-bandwidth limit of chip-scale spectrometry by harnessing a dispersion-engineered photonic molecule. Light: Science & Applications **12**, 64 (2023). URL <https://www.nature.com/articles/s41377-023-01102-9>.
- [35] Yang, Z. et al. Single-nanowire spectrometers. Science **365**, 1017–1020 (2019). URL <https://www.science.org/doi/10.1126/science.aax8814>.
- [36] Yao, C. et al. Broadband picometer-scale resolution on-chip spectrometer with reconfigurable photonics. Light: Science & Applications **12**, 156 (2023). URL <https://www.nature.com/articles/s41377-023-01195-2>.
- [37] Zhang, J., Cheng, Z., Dong, J. & Zhang, X. Cascaded nanobeam spectrometer with high resolution and scalability. Optica **9**, 517–521 (2022). URL <https://opg.optica.org/optica/abstract.cfm?uri=optica-9-5-517>.
- [38] Zhang, Z. et al. Integrated scanning spectrometer with a tunable micro-ring resonator and an arrayed waveguide grating. Photonics Research **10**, A74–A81 (2022). URL <https://opg.optica.org/prj/abstract.cfm?uri=prj-10-5-A74>.
- [39] Zhang, Y. et al. Miniature computational spectrometer with a plasmonic nanoparticles-incavity microfilter array. Nature Communications **15**, 3807 (2024). URL <https://www.nature.com/articles/s41467-024-47487-y>.

- [40] Zhang, L. et al. Ultrahigh-resolution on-chip spectrometer with silicon photonic resonators. Opto-Electronic Advances **5**, 210100–210109 (2022). URL <http://www.chinaoceanengin.cn/en/article/doi/10.29026/oea.2022.210100>.
- [41] Zheng, S. N. et al. Microring resonator-assisted Fourier transform spectrometer with enhanced resolution and large bandwidth in single chip solution. Nature Communications **10**, 2349 (2019). URL <https://www.nature.com/articles/s41467-019-10282-1>. Publisher: Nature Publishing Group.
- [42] Zhu, A. Y. et al. Ultra-compact visible chiral spectrometer with meta-lenses. APL Photonics **2** (2017). URL <https://pubs.aip.org/aip/app/article/2/3/036103/122962/Ultra-compact-visible-chiral-spectrometer-with>. Publisher: AIP Publishing.

Response to Reviewers:
Achieving Sub-pm Wavelength Regression via Minimum-Phase in a
Single-Stream Photonic IC

Hector A. Rubio Rivera^{1,*}, Lilian Neim¹, Venkatesh Deendayalan¹, Stefan Preble¹

¹Electrical and Microelectronic Engineering, Rochester Institute of Technology, NY, USA

*Corresponding author: hr8392@rit.edu

Reviewers' Comments and Author's Response

Dear Reviewers,

We thank you for the evaluation of our manuscript, "Achieving Sub-pm Wavelength Regression via Minimum-Phase in a Single-Stream Photonic IC," and for the constructive feedback provided during this review process. In addition, we are grateful for the timely response to our first revision round and the positive feedback on the second revision round.

In this revised response, we address the remaining points raised by Reviewer 1 concerning scalability, signal-to-noise ratio constraints, and operational bandwidth limitations. Specifically, we have (i) added a new supplemental section presenting a system-level signal-to-noise ratio analysis that quantifies the bounds on the maximum allowable delay length for our minimum-phase multi-tap interferometer architecture, and (ii) clarified the origin of the demonstrated 10 nm operational bandwidth by explicitly distinguishing intentional architectural design choices from practical constraints imposed by the directional couplers/splitters. Corresponding clarifications have also been incorporated into the main manuscript text. We have additionally addressed the inconsistencies noted by Reviewer 3 and are grateful to Reviewer 2 for their positive assessment of the revised manuscript. We believe these revisions enhance the clarity of the work and better position the contribution within the broader context of photonic integrated wavemeters and phase-retrieval-based wavelength regression.

Thank you for your time and consideration. We look forward to your guidance on the next steps in the review process.

Reviewer 1

Remarks to the Author:

I would like to thank the authors for their detailed responses and the significant revisions made to the manuscript. Most of my previous concerns regarding the methodology and the positioning of the work have been addressed. However, there are two remaining points regarding the scalability and the operational constraints of the proposed architecture that require further clarification before the manuscript is suitable for publication.

Response:

We thank the reviewer for their insightful evaluation of our work and for helping us improve the quality of our manuscript. We have addressed their thoughtful points below.

Reviewer Comment 1:

The authors demonstrate that resolution scales with the length and number of delay taps. However, the power of the reference arm must remain dominant to satisfy the stability of the Hilbert transform phase retrieval. As the number of delay taps increases, or as path lengths extend (leading to higher propagation losses), the power budget for each individual interferometric component is naturally squeezed. This inevitably constrains the system's dynamic range. It is recommended to discuss how the input power and the system's Signal-to-Noise Ratio impact the accuracy of the wavelength regression.

Response:

We thank Reviewer 1 for their insightful comment regarding the interplay between delay length, reference-arm dominance, and signal-to-noise ratio (SNR) in the proposed architecture. The methodology of our approach hinges on the ability to properly resolve the delay lines in the interferogram Fourier space, as demonstrated in Fig. S4 of the Supplemental. As such, we studied the SNR budget of a given n th tap within our multi-tap interferometer architecture with the intention of solving for the maximum allowable length that can be recovered from this Fourier space to yield a practical SNR. We added a new supplemental section titled “VI. Signal-to-Noise Ratio Analysis for Multi-Tap Interferometer”, and cited this in the main paper in Section III with the following sentence: (see Section VI of the Supplement for a detailed system-level SNR analysis). In this section, we address the trade-off pointed out by the Reviewer by deriving an expression for the per-tap SNR, including both amplitude noise introduced by the detection chain and phase noise arising from thermo-optic and environmental fluctuations in the photonic circuit.

We found that using detector noise values from the manufacturer's datasheet and a

representative SNR threshold ($\text{SNR} \geq 10$ required to reliably resolve a delay line in Fourier space), we find that the maximum recoverable delay length under detector-noise-limited operation lies in the range of approximately $\sim 50\text{--}70$ cm (with a typical 1dB/cm propagation loss) for the experimental system considered. When phase noise is taken into account, thermo-optic fluctuations further limit the usable delay length to the $\sim 2\text{--}20$ cm range under typical ambient conditions. These bounds quantify how extremely attenuated tap powers ultimately degrade the resolvability of the corresponding Fourier components, thereby limiting their usefulness in our wavelength regression algorithm.

Importantly, increasing the input optical power does not arbitrarily relax these bounds. While higher optical power can improve SNR, it is ultimately limited by nonlinear optical effects in the waveguides. These nonlinearities set an upper bound on usable launch power and therefore constrain the extent to which SNR can be improved purely by power scaling. However, this nonlinear limitation is not intrinsic to the proposed phase-based wavelength regression method. Rather, it is a platform-dependent constraint that may be mitigated through alternative waveguide geometries (e.g., larger mode areas), material platforms with lower nonlinear coefficients (such as silicon nitride or doped silica), or improved thermal management and packaging. Consequently, the maximum achievable dynamic range and delay length reflect a combined optimization over waveguide design, material choice, detector noise, and phase stability.

Finally, SNR is not the sole constraint on the maximum usable delay. Extending the path length also decreases the free spectral range (FSR), placing increasing demands on the wavelength stepping resolution of the calibration laser. For delay lengths approaching the SNR-limited bound, the FSR reaches the picometer scale, requiring laser tuning granularity that may exceed the capabilities of practical sweep sources. Consequently, although increasing the delay length improves the theoretical wavelength sensitivity, it simultaneously imposes stricter requirements on both optical SNR and calibration spectral resolution. Taken together, these considerations show that the practical performance of the proposed wavelength regression method is determined not by the phase-retrieval algorithm itself, but by system-level constraints. The optimal choice of delay lengths and number of taps is therefore application dependent and must balance achievable SNR, dynamic range, and calibration capabilities rather than relying solely on increasing optical path length.

Reviewer Comment 2:

The manuscript notes that the tunable laser used in the experimental setup is capable of a wide sweep from 1500 nm to 1600 nm. However, the characterized operational range of the Photonic IC is limited to a 10 nm window. It is currently unclear whether this limitation is a fundamental design choice or a practical constraint.

Response:

We thank Reviewer 1 for their comment regarding the bandwidth limitation of our demonstration. We note that this limitation arises from a combination of intentional design choices and practical constraints of the fabricated device. The multi- ΔL photonic IC incorporates two distinct wavemeter architectures: a 4-tap and an 8-tap design. The 4-tap architecture was intentionally designed for a 10nm operational bandwidth, corresponding to the shortest delay line (ΔL) that maintains a uniquely determined phase over this spectral range. In contrast, the 8-tap architecture was designed to support a much broader bandwidth (100nm). However, as shown in Supplemental Fig. S4, the 8-tap design does not satisfy the necessary conditions for accurate signal reconstruction. Specifically, the selected delay line values lead to significant spectral leakage in Fourier space, resulting in poorly resolved delay components. Consequently, the reconstructed models derived from the 8-tap design are unable to reliably reproduce the original photodetected signal, unlike the 4-tap design, which consistently enables accurate reconstruction.

In addition to these architectural considerations, the bandwidth of the directional couplers imposes a practical constraint. The couplers violate the minimum-phase condition outside a narrow wavelength range. Using our reconstruction-based extraction method, we measure the coupler bandwidth to be approximately 10nm centered near 1550nm (Supplemental Fig. S4). This limitation ultimately defines the effective operational bandwidth of the device. This can be overcome with couplers that operate over a large bandwidth range, such as adiabatic couplers[1].

Although the tunable laser source allows sweeping from 1500nm to 1600nm, this wide tuning range is used for characterization rather than intended operation. Sweeping beyond the nominal design bandwidth enables us to probe the phase retrieval process over a broader spectral range, and in particular, to extract the bandwidth of the directional couplers. The operational bandwidth of the photonic IC is therefore constrained to ~ 10 nm by both the delay-line design and the coupler response, rather than by the laser source itself. We added the following sentence in section IV.B to clarify that the demonstrated 10 [nm] bandwidth is both a design choice and a practical limitation of the used directional couplers: *Although the tunable laser supports sweeping from 1500–1600 nm, the demonstrated 10 nm operational bandwidth of the photonic IC is set by intentional delay-line design choices and by practical limitations of the directional couplers, rather*

than by the laser source itself.

Reviewer 2

Remarks to the Author:

The authors have provided a comprehensive response to all comments and addressed all technical concerns. Their revisions enhance the manuscript's quality, resolving key issues and strengthening their contribution to the field of photonic integrated circuits (PICs).

1. The authors clearly distinguished their approach from the prior work proposed by Xu et al., demonstrating that a sequential tap design would require 100 delay lines for the same functionality as in the manuscript.
2. The manuscript introduces the work of Mecozzi et al., which provides the basic theory to link the minimum phase with the phase recovery from intensity measurements.
3. The manuscript has been revised to optimize the proposed wavelength regression algorithm in Figure 2 and enhance readability.

Response:

We thank the reviewer for their insightful evaluation of our work and for helping us enhance the manuscript quality.

Reviewer 3

Remarks to the Author:

I would like to thank the authors for addressing my comments. I believe the manuscript is stronger now and I do not have any further comments except for the following: On page 8 (Methods) there is inconsistency in the loss per facet (7 dB vs 9 dB). Please fix that.

Response:

We thank the reviewer for their evaluation of our work and for helping us strengthen the manuscript quality. We assure the reviewer that there is no inconsistency between the loss per facet, as the material platform for each design is different. The single ΔL design is fabricated on Si by AIM Photonics, while the multi- ΔL design was fabricated on SiN by Applied Nanotools. Hence, the different loss per facet figures.

References

- [1] Cabanillas, F. & Maria, J. Rapid adiabatic devices enabling integrated electronic-photonic quantum systems on chip (2022). URL <https://hdl.handle.net/2144/44749>.